# Virulent coliphages in 1-year-old children fecal samples are fewer, but more infectious than temperate coliphages

Aurélie Mathieu[1], Moïra Dion[2,3], Ling Deng[4], Denise Tremblay[3,5], Elisabeth Moncaut[1], Shiraz A. Shah [6], Jakob Stokholm[6], Karen A. Krogfelt [7], Susanne Schjørring[7], Hans Bisgaard [6], Dennis S. Nielsen [4], Sylvain Moineau [2,3,5] & Marie-Agnès Petit [1]*

Bacteriophages constitute an important part of the human gut microbiota, but their impact on this community is largely unknown. Here, we cultivate temperate phages produced by 900 *E. coli* strains isolated from 648 fecal samples from 1-year-old children and obtain coliphages directly from the viral fraction of the same fecal samples. We find that 63% of strains hosted phages, while 24% of the viromes contain phages targeting *E. coli*. 150 of these phages, half recovered from strain supernatants, half from virome (73% temperate and 27% virulent) were tested for their host range on 75 *E. coli* strains isolated from the same cohort. Temperate phages barely infected the gut strains, whereas virulent phages killed up to 68% of them. We conclude that in fecal samples from children, temperate coliphages dominate, while virulent ones have greater infectivity and broader host range, likely playing a role in gut microbiota dynamics.

[1] Université Paris-Saclay, INRAE, AgroParisTech, Micalis Institute, 78350 Jouy-en-Josas, France. [2] Département de biochimie, de microbiologie, et de bio-informatique, Faculté des sciences et de génie, Université Laval, Québec City, QC G1V 0A6, Canada. [3] Groupe de recherche en écologie buccale, Faculté de médecine dentaire, Université Laval, Québec City, QC G1V 0A6, Canada. [4] Department of Food Science, University of Copenhagen, Copenhagen, Denmark. [5] Félix d'Hérelle Reference Center for Bacterial Viruses, Faculté de médecine dentaire, Université Laval, Québec City, QC G1V 0A6, Canada. [6] Copenhagen Prospective Studies on Asthma in Childhood, Copenhagen University Hospital, Herlev-Gentofte, Ledreborg Allé 34, DK-2820 Gentofte, Denmark. [7] Department of Bacteria, Parasites and Fungi, Statens Serum Institut, Artillerivej5, 2300S Copenhagen, Denmark. *email: marie-agnes.petit@inra.fr

Viral particles abound in the human intestinal microbiota, with $10^9$ to $10^{10}$ estimated virus-like particles (VLPs) per gram of feces[1–3]. Metagenomics has repeatedly demonstrated the great richness in this viral world, which is largely dominated by bacteriophages (or phages)[4–8]. Despite the richness, there is low intra-sample diversity (when evenness is taken into account) because a few groups of abundant phages appear to dominate the ecosystem[9,10] of each individual, and this phageome seems relatively stable over time[5,11]. In contrast, inter-sample diversity is large, and each individual essentially has its own set of phages. Yet, it is estimated that 20–25 phages, including crAssphage[9], are abundant and shared by more than 20% of adults[10]. Since phages have been established as important components of the human microbiota, it is necessary to evaluate their impact in this environment[12].

Lysogeny is frequent in bacterial strains, with 46% of completely sequenced bacterial genomes hosting at least one prophage[13], and seems even more frequent in intestinal bacteria[14]. Mouse studies also reported that 66% of lysogens were actively producing viral particles[14]. In humans as well, the first intestinal virome studies underlined that a large fraction of VLPs might correspond to temperate phages[6,8]. Virulent phages are also present in the human digestive tract. A 1983-study found phages targeting E. coli (coliphages) at levels higher than $10^5$ per gram of feces in 1.6% of the healthy subjects, and in 14% of diarrheal patients[15]. The increased occurrence of coliphages in diarrheal patients was likely a reflection of a concomitant increase in the presence of their E. coli hosts. Among these coliphages, the proportion of virulent coliphages was 10% in the fecal samples of healthy adults, and 24% in the diarrhea samples. These figures suggest first, a dominance of temperate coliphages compared with virulent coliphages in a balanced microbiota, and second, a tendency for the virulent phages to expand in "dysbiotic" contexts such as during the occurrence of diarrhea. Interestingly, a similar study conducted approximately 20 years later examined the fecal samples of 140 Bengal children with acute gastro-enteritis and the authors succeeded in isolating coliphages from 27% of the samples, and 95% of them were virulent[16]. These findings support the view that a diarrheal context correlates with virulent phage development.

Beyond coliphages, a suggestion that phages may play a role in microbiota stability came from a large virome study, which detected an increase in virome richness in Crohn's disease patients, compared to healthy subjects[17]. Furthermore, a re-analysis of this dataset focusing on the temperate phages associated with Faecalibacterium prausnitzii, a dominant bacterial member of the human microbiota which is depleted in Crohn's disease patients, suggested a negative correlation between bacterial and viral abundances for this species[18]. Such a situation may originate from massive prophage induction in a dysbiotic context. The impact of oral administration of either defined virulent phages or complex viromes on the intestinal microbiota has also been investigated, and has two possible outcomes. One is that phage propagate without impacting the bacterial population, which remains globally phage sensitive, in a manner of co-existence[19,20]. In the second possible outcome, phages affect the bacterial population in a transient way, which then recovers to initial concentrations[21] either via strain replacement[21] or via evolution towards phage resistance[22]. These studies pave the way to understand phage impact(s) on intestinal communities.

However, at present we are far from having a well-defined view of the respective contributions of temperate and virulent phages in the mammalian gut. We undertook such a study with E. coli, one of the first bacterial species to colonize an infant's gut, which still represents more than 1% of all species of the gut microbiota in many children at age one year. We isolated 150 E. coli phage

and analyzed their host range on 75 E. coli strains isolated from the same ecosystems, starting from 648 fecal samples of a one-year-old children cohort. We show that temperate phages outnumber virulent ones in fecal samples, and that most E. coli strains isolated from the same cohort are resistant to temperate phages but sensitive (for 11–60% of them) to virulent phages. Therefore, this study generates a contrasting picture, where temperate coliphages are relatively abundant but remarkably innocuous to bacteria found in one-year-old children, and virulent phages are sub-dominant but may kill their host upon encounter.

## Results

**E. coli strains from fecal samples are mostly lysogens.** To study intestinal E. coli phage–host interactions, we isolated phages from an unselected prospective cohort of 1-year-old children. Using 648 fecal samples, two parallel investigations were carried out. During the first (Fig. 1, left side), we analyzed the temperate phages hosted in 900 E. coli strains previously isolated from these same fecal samples (0–5 isolates per sample, see[23]). These E. coli strains were searched for the presence of virions spontaneously induced in in vitro culture supernatants by plaquing them on two indicator strains.

Two laboratory E. coli strains were chosen for phage isolation based on their high phage sensitivity. Strain MAC1403 (MG1655 hsdR::kanR) is a K12 derivative (phylogroup A) devoid of active

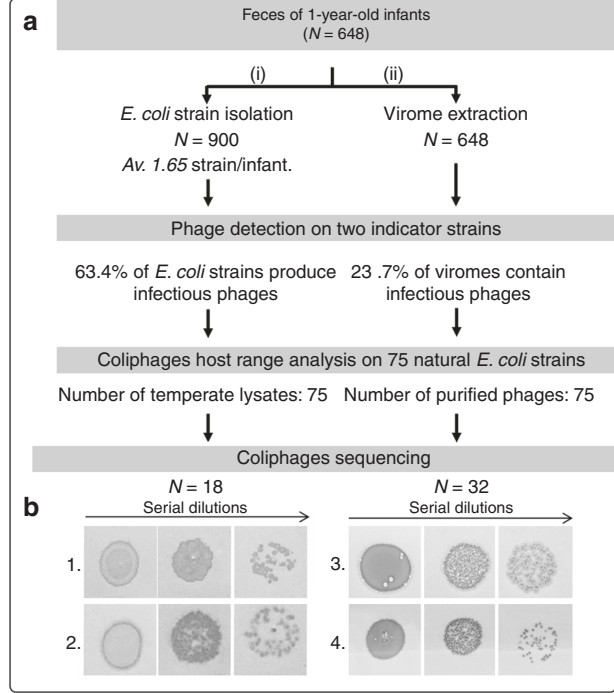

**Fig. 1 Coliphages from 1-year-old infant fecal microbiota: overview of the study. a** Fecal samples from 648 1-year -old infants were collected and used (i) to isolate E. coli strains and (ii) to extract their viromes. Next, two laboratory strains were chosen to culture coliphages from these two sources. Subsequently, using a subset of 75 coliphages from each source, the host-range was determined based on 75 E. coli isolates from the same cohort. Finally, the genomes of 18 temperate coliphages isolated from E. coli strains and 32 coliphages isolated in viromes were sequenced. **b** Examples of temperate/virulent phenotypes of the coliphages isolated from viromes. Various dilutions of coliphages amplified on either of the two indicator strains were spotted on soft agar plates inoculated with an E. coli indicator strain. Confluent growth and isolated plaques of two phages making turbid (1 and 2) and clear (3 and 4) plaques are shown.

prophage and with no DNA restriction activity. It has a lipopolysaccharide (LPS) with a K12-type outer-core component, and is devoid of O-glycosylation (rough phenotype). Strain C also belongs to phylogroup A, is restrictionless, has an LPS devoid of O-glycosylation but has an R1 type outer-core[24]. The *E. coli* C strain has long been recognized as an efficient indicator strain[15]. Due to size limitations, no strains with an F pilus were used, thereby excluding detection of phages that need such pili for adsorption (members of the *Inoviridae* and *Leviviridae* families)[15].

For the 900 natural *E. coli* isolates from the fecal samples of 1-year-old children, filtered supernatants of 96-well plate cultures were serially diluted and plated on the two indicator strains. The observation of isolated plaques was considered proof of the presence of temperate phage(s) in the supernatant (Supplementary Fig. 1B). The detection limit was $10^2$ pfu/mL of supernatant. Remarkably, 63.4% of the strains produced virions that were able to form plaques on either or both of our indicator strains. In all cases, confluent lysis spots and isolated plaques exhibited a turbidity zone typical of temperate phages (see two examples, Fig. 1b, left). Supernatant titers ranged between $10^2$ and >$10^6$ pfu/mL (Supplementary Fig. 1C).

We conclude that at least 63% of *E. coli* isolates from the infant gut are lysogens. Among the phages isolated, 75 were selected for further analysis. These phages were chosen based on their different plaque characteristics (size, turbidity), with a maximum of one phage per initial fecal sample.

**Viromes contain both temperate and virulent coliphages.** The second route to isolate coliphages (Fig. 1, right side), consisted in growing them directly from fecal sample filtrates. Viromes were fractionated from 0.15 g of feces, using a gentle method based on fecal sample dilution, filtration, and concentration through ultra-filtration. From such viromes, plaque isolation was then performed, on the same two indicator strains used for strain supernatants.

Coliphages were isolated from viromes by direct spotting (10 µL) on lawns of MAC1403 and C strains, without an enrichment step. Among the 648 virome samples, 23.7% contained coliphages (>500 pfu/g of feces, Supplementary Table 1). Fecal samples with coliphages isolated had a significantly higher proportion of *E. coli* OTUs compared to coliphage negative samples (Mann–Whitney two-tailed test, $P < 0.0001$, Supplementary Fig. 1D).

Interestingly, some of these phages produced clear plaques, suggesting that virulent phages had been isolated (Fig. 1b, right

side). Among the first 75 isolated plaques, which were studied further, 27% were virulent. We noticed that fecal samples positive for virulent coliphage contained slightly higher *E. coli* OTU proportions (Fig. 2a), and had significantly higher coliphage titers ($3.1 \times 10^4$ PFU/g of feces for virulent, compared to $2.8 \times 10^3$ for temperate phages, Fig. 2b). When the MOI of the coliphages was plotted as a function of *E. coli* concentration (using an absolute bacterial concentration of $10^{11}$/g of feces to convert *E. coli* OTU ratio into concentrations), a general trend of lower MOI for higher bacterial concentration was observed (Fig. 2c). Such a tendency has already been reported for several marine bacterial genera and the combined population of their phage parasites[25]. It was proposed that temperate phages shift from lysis cycles to lysogeny at high bacterial concentrations (the "piggyback the winner" hypothesis[26]). Interestingly, here we see a similar behavior for both temperate and virulent phages. Alternatively to the piggyback the winner hypothesis, it is possible that with phage adsorption to the bacterial surface being more efficient at high bacterial concentrations, less virions are collected in supernatants, regardless of the phage lifestyle.

The titers of coliphages in viromes may be underestimated, due to purification procedure and phage storage. Indeed, our phage spiking assays showed that the virome purification procedure reduced infectivity by a factor of 2.5, in average[27]. With respect to storage, we observed that phage particles remained stable upon storage at −80 °C in fecal samples. Later on along the purification process, for logistical reasons, there was a 3 weeks' delay between virome isolation and growth analysis, which may result in phage decay. Indeed, a stability assay at 4 °C with 12 temperate and 11 virulent coliphages crude lysates showed that all temperate coliphages but one suffered a 1 to 2 log decay in titer over 35 days (Supplementary Fig. 2), whereas virulent phages were slightly more stable over the same period.

Taken altogether, the viromes of healthy children contain both temperate and virulent coliphages, 75 of which were further purified and analyzed in greater detail.

**Temperate and virulent phages have contrasted host range.** Next, we tested the capacity of the selected 150 phages (75 spontaneously induced and 75 isolated from viromes) to replicate on a panel of 75 *E. coli* strains. A maximum of one strain per child was chosen, half of which were proven lysogens (i.e. strains for which phages could be detected in supernatants). We

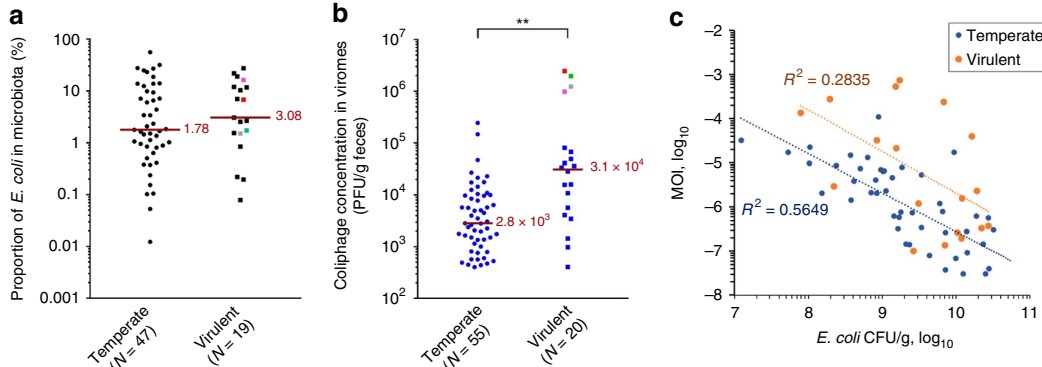

**Fig. 2 *E. coli* and coliphage levels in microbiota, segregated by phage lifestyle. a** A slight increase of the fraction of *E. coli* to total microbiota OTU is found among samples containing virulent coliphages, compared to those where temperate phages were found (not statistically significant, Mann–Whitney test, $P = 0.39$). **b** The overall coliphages titer in the fecal sample is significantly higher for virulent, compared to temperate phages (**, $P = 0.0002$). In both first panels, red line = median value, four colored points = top coliphages titers. **c** The multiplicity of infection (MOI) correlates negatively with *E. coli* concentration in the fecal samples, both for virulent and temperate phage containing samples ($R^2$, coefficient of determination of the power law). Source data are provided as a Source Data file.

made sure that none of the selected *E. coli* strains originated from a child from which phages to be tested had been isolated (this was also tested, later, see below). The *E. coli* species is diverse, and seven phylogroups are distinguished, with different overall properties. Clades A and B2 are the most prevalent in humans, with A strains being mostly commensals while most extra-intestinal pathogens belong to the B2 clade[28]. A PCR typing was performed on the 75 *E. coli* strains in order to determine their phylogroup[29]. We observed a clear dominance of the B2 type (41.3% of strains), and a paucity of the A type (5%). Clades E, B1 and C had intermediate levels (11–19%). Interestingly, the B2 clade was significantly enriched in lysogens, while B1 and A were significantly depleted in lysogens (Supplementary Fig. 3A).

Among the 75 temperate phages obtained from *E. coli* culture supernatants, the 18 selected to be sequenced were purified on one of the two indicator strains. The remaining 57-phage stocks were left as crude bacterial supernatants, to mimic natural infection conditions in terms of phage diversity and exposure to multiple DNA modification systems. A remarkably low level of infectivity was observed for the whole 75-phage panel (Fig. 3a). At most, 8% of the strains were lysed by a temperate phage. Only 2 of the 75 strains were more phage sensitive (i.e. >20% of the phages could infect them).

Next, we tested the panel of 75 phages isolated from viromes and purified on either one of the two indicator strains against the panel of 75 *E. coli* isolates (Fig. 3b). The 55 temperate phages behaved similarly to the previous temperate phage set, with at most 9% of the strains lysed. In marked contrast, the 20 virulent phages lysed between 11% and 68% of the strains. Interestingly, four virulent phage lysed at least 60% of all strains tested (three rV5 and one phAPEC8, see below). Lastly, we chose 16 known virulent coliphages from the Felix d'Hérelle Reference Center for Bacterial Viruses as a reference set, and tested them against the same panel of 75 *E. coli* strains (Fig. 3c). Their infectivity fell between the temperate and the virulent phage sets, as between 0% and 37% of the strains were lysed by this reference phage set.

We concluded that the overall infectivity of the three phage groups examined (temperate, virulent, and reference virulent) are clearly distinct (Supplementary Fig. 3B). The virulent coliphages isolated from the infant gut were the most infectious towards the strains of this niche, followed by the reference virulent phages, and then the temperate phages. The variations in phage sensitivity could not be related to the lysogeny status of the tested strains, as lysogens and non-lysogens were equally resistant to phages (Supplementary Fig. 3C). The *E. coli* phylogroups also behaved similarly (Supplementary Fig. 3D).

To complement the global phage–bacteria interaction matrix, we tested each phage on a few *E. coli* strains isolated from the same fecal sample (Fig. 3d). The overall trend observed for the interaction matrix was conserved. All but 3 of the 130 temperate phages were unable to replicate on the strains of their ancestral ecosystem, whereas 9 of the 20 virulent phages replicated on at least one strain of the niche from which it was initially isolated.

We then used the interaction matrix to examine whether phage–host interactions were nested or modular. Earlier studies found that at the species level, phage–host interactions were nested, with a gradient of interactions ranging from generalist to specialist phages[30]. However, at larger taxonomic and/or geographic scales, interactions started to become modular, with some phages and hosts interacting more with each other than across their module[14,31]. With *E. coli* being a large and multifaceted species, either outcome might be expected. With our interactions matrix, we observed an intermediate situation where no clear modules emerged, nor did the interactions fit with a strict nested situation (Supplementary Fig. 4a, b).

**Some temperate phages may have crossed genera**. We sequenced the genomes of 50 phages of our collection, 18 temperate from isolated strain supernatants, 17 temperate from viromes, and 15 virulent from viromes (labeled in red in Fig. 2), then assembled and annotated them. All genome information is summarized Fig. 4. To our surprise, relatives to all but two coliphages (Fraca and Evi) had been characterized previously. Phages were named after their closest relatives in databases (usually at the species/genus level), followed by virome number ("ev" suffix) or strain number. As already reported, we found that virulent phages did not necessarily grow as clear plaques on all infected strains[32]. Since all those with sequenced genomes made clear plaques on at least three *E. coli* strains, we used this rule to declare the remaining unsequenced phages in Fig. 3 as virulent.

The eighteen genomes of the temperate phages isolated from *E. coli* strains grouped into three distinct clusters (Supplementary Fig. 5). The most represented were Lambdavirus (10 of the 18) closely related to the siphophage Lambda, except for a mEP460. The next largest cluster comprised seven P2virus, closely related to the myophage P2, except for a P88. The last phage was a myophage Mu. This distribution is congruent with bioinformatics analyses showing that lambda and P2 are among the most prevalent prophages in *E. coli* genomes[33].

These phage genomes were then compared to those of 17 temperate phages isolated from viromes. They grouped into four clusters (Supplementary Fig. 6), revealing again the prevalence of Lambdavirus (11 out of 17 genomes). Four isolates were related to the *Cronobacter sakazakii* temperate phage ESSI-2 (31% coverage, 73% nucleotide identity on average, see Supplementary Fig. 6), which is a member of the *Myoviridae* family[34], placed taxonomically within the P2virus genus. A search for ESSI-2 prophages in Enterobacteriaceae with megablast revealed its presence also in *Enterobacter* (7 of 95 genomes), *Citrobacter* (8 of 52 genomes), and *Salmonella* (43 of 459 genomes). In contrast, they were scarce in *E. coli* (1 of 10,943 genomes). The prophages with the nearest percentage of identity to our *E. coli* infecting ESSI-2 were from *Escherichia alberti* and a *Citrobacter* species (96–95% identity, 81–92% query cover, respectively). Therefore, these ESSI-2 phages infecting *E. coli* may originate from a different bacterial species, or genus.

The final two temperate phages for which the genome was sequenced, Fraca and Evi, were singletons with no close relative sharing more than 30% of their length in the NCBI viral nucleotide database. The genome of phage Fraca is 43,664 bp long, and analysis of its neck module using Virfam[35] classified this phage together with the siphophage HK97 (Type 1, cluster 3). A few prophages related to Fraca were found in *Enterobacter*, suggesting that Fraca may have crossed genera to infect *E. coli* in our assay, as in the case of ESSI-2. The second singleton, Evi (40,019 bp), shared a third of its deduced proteome with the *E. coli* temperate phage Phi467 (structural module). This module was frequently encountered in prophages of *E. coli*. Using Virfam classification, its neck module was also classified next to the siphophage HK97 (Type 1, cluster 3), but Fraca and Evi barely shared nucleotide similarity (5% coverage, 77% identity).

**rV5 related virulent phages are most infectious**. Among the 15 sequenced genomes from virulent phages (Fig. 1 and Supplementary Fig. 6), five were closely related to the myophage rV5 of the *Vequintavirinae* subfamily, and one of them infected up to 68% of the tested *E. coli* strains (range 24–68%, average 52%). In addition, we isolated a phAPEC8-like phage, which is distantly related to rV5[36] (18% shared proteins), that infected 62% of all strains tested. This underlines the remarkable capacity of

# ARTICLE

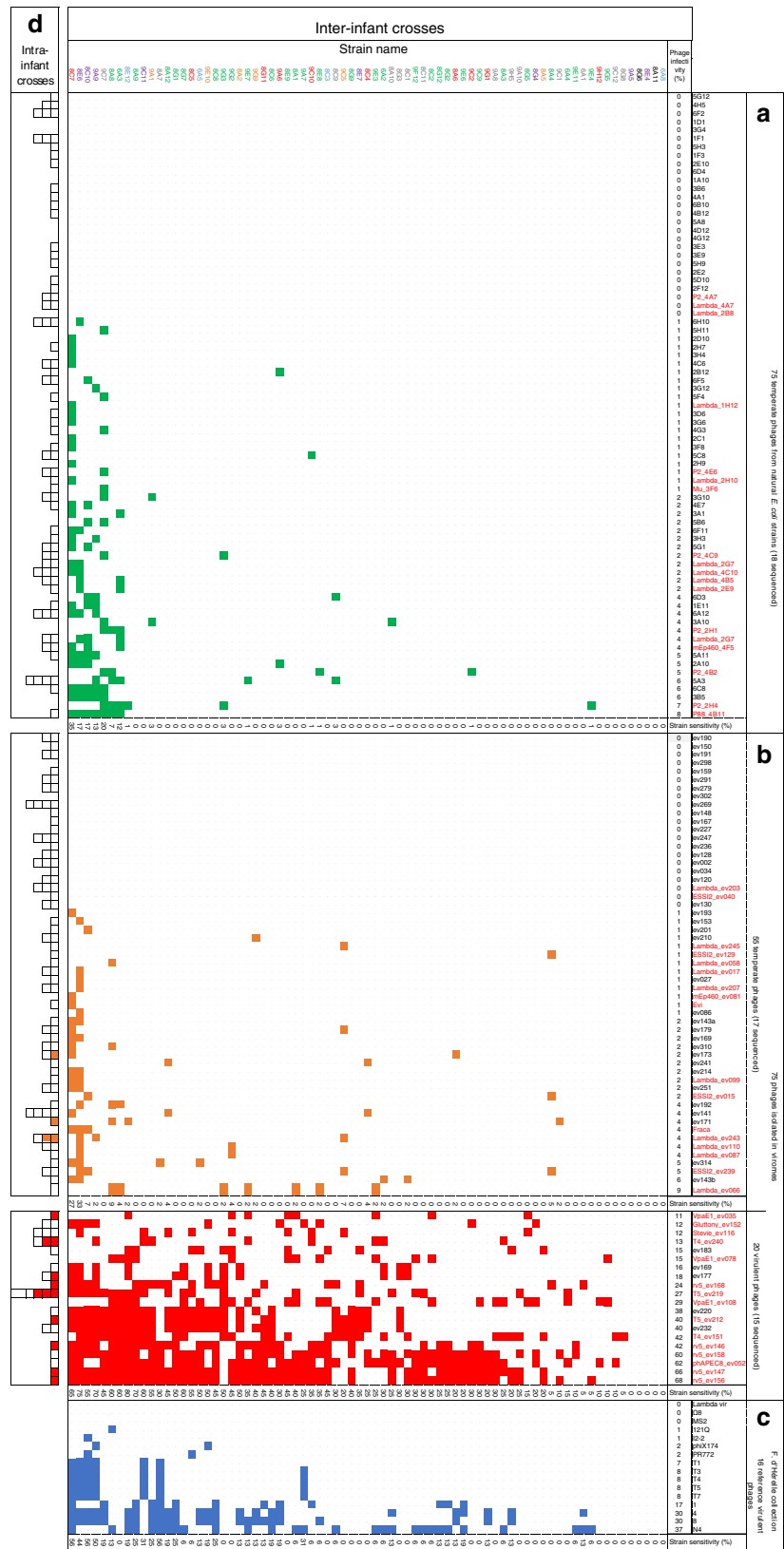

members of the *Vequintavirinae* subfamily to adapt and infect various *E. coli* strains isolated from the infant gut.

The next most infectious phages were related to the siphophage T5 (two isolates) and myophage T4 (two isolates). Of note, the host range was distinct within each phage pair. The two T5-like phages infected 27 and 40% of the strains (see below), while the two T4-like phages infected 13 and 42% of the strains.

A set of three myophages were homologous to VpaE1 (88,403 bp), a member of the FelixO1virus genus, which can replicate over a wide range of temperatures (from 9 to 45 °C), and has

**Fig. 3 Phage impact on natural _E. coli_ strains.** Bacterial lawns of 75 _E. coli_ isolates from infant fecal samples were infected with 166 coliphage spots. The left column shows strain names with color-coded phylogroups: A in blue, B1 in red, B2 in green, C in purple, D in orange, E in gray and unclassified _Escherichia_ clades in black. Upper row: phage names, split into 3 categories (panels (**a**)–(**c**)). Each square of the matrix corresponds to a phage–strain interaction (white: no phage growth, color: phage growth, green, temperate isolated from a lysogen, orange, temperate isolated from a virome, red, virulent isolated from virome, blue, virulent from d'Hérelle collection). In panels (**a**), (**b**), all crosses are inter-infant (i.e. phage and strain never come from the same infant). The "phage infectivity" line sums up the percentage of strains killed by each phage. The "strain sensitivity" column sums up the percentage of phages in each category infecting this strain. **a** 75 temperate phages derived from fecal _E. coli_ strains supernatants, either as crude lysates (57 of them, black entry) or as plaque purified and amplified on an indicator strain (18, in red, completely sequenced). **b** 75 coliphages found in viromes, plaque purified and amplified on an indicator strain. To determine the temperate or virulent phenotype, see text. The genome of phages labeled in red were completely sequenced. **c** 16 reference virulent coliphages obtained from the Felix d'Hérelle Reference Center for Bacterial Viruses (www.phage.ulaval.ca). **d** Intra-infant crosses (infant coliphages tested on the strains originating from the same fecal sample). For phages in panel (**a**), the _E. coli_ strain from which the phage was collected is excluded.

interesting host range properties[37]. For example, its receptor is composed of a truncated core LPS. It was suggested that phage VpaE1 preferentially targets _E. coli_ mutants whose LPS cores are truncated to resist infection by other phages such as T4. However, a close relative of VpaE1, called Alf5, was previously isolated and infects strain K12, which has a complete core LPS[38]. Phage Alf5 encodes different alleles of the tail fiber proteins involved in host recognition. We compared the newly isolated VpaE1 phages to these two phages, and found that they had a tail fiber more closely related to VpaE1_gp74 (86% to 97% identity) than to its counterpart in Afl5 (74–75% identity). This may explain the limited host range by our VpaE1-like isolates (11%, 18%, and 29% of the tested strains).

Finally, the two remaining virulent phages were homologous to Stevie (49,816 bp) and Gluttony (44,513 bp), and infected 12–18% of the strains. Gluttony is an _E. coli_ bacteriophage isolated from the microbiota of a female bladder for which little is known. Its neck module classification with Virfam places Gluttony not far from PA73, in type 1, cluster 5. Stevie is related to T1 and was originally isolated from _Citrobacter_[39].

Overall, the virulent phages isolated from our viromes were more diverse than temperate phages, with some exhibiting remarkably high infective powers.

**Accessory genes of the intestinal coliphage genomes.** Three categories of genes are particularly interesting to detect in phage genomes, because they reveal how phages interact with their host, either by increasing its pathogenicity (virulence factors), or more generally its fitness (morons), or by boosting its metabolism so that the phage can complete its lytic cycles (auxiliary metabolic genes, AMG). We screened virulence factors using the VFDB reference database and the VFanalyzer tool, with _Escherichia_ as a reference host[40]: 18 of the 50 sequenced phages encoded such proteins (reported in Fig. 4), all but one being temperate Lambda phages. The most abundant was _sitABCD_, an operon encoding an iron and/or manganese transporter, which resided on 70% of the lambda phages. Its presence may suggest that iron and/or manganese are limiting factors for _E. coli_ in the intestinal microbiota of infants. In all cases, a _fur_ box was detected upstream of the _sit_ operon, suggesting that its expression is fully integrated into the host Fur regulon (Supplementary Fig. 7). Three effectors were also detected: a SopA outer membrane protease (found twice), which is required for the proper displacement of intracellular Shigella on actin filaments, and is also injected into mammalian cells through the type III secretion system (TTSS), and interferes with E3 ubiquitin ligase[41–43]. In addition EspM2 and EspV effectors were found in Lambda_2B8 and Evi, respectively. These effectors are usually encountered in attaching-effacing enteropathogens. Upon translocation by TTSS, EspM2 modulates the actin dynamic through interaction with the RhoA GTPase[44], while EspV drastically changes cell shape[45]. Even though effector

genes are the apanage of pathogens, their frequent presence on prophages[46] allows them to shuttle between commensal and pathogenic strains. Hence, effector detection does not necessarily imply that pathogenic _E. coli_ were present in the corresponding samples. _De facto_, the EspM2 and EspV encoding phages both originated from the same diarrheic fecal sample, but the SopA encoding phages were isolated from two non-diarrheic fecal samples.

A search for antibiotic resistance genes using the Resfam[47] core profiles revealed a small efflux pump EmrE in two lambda genomes. Such pumps permit to eliminate tetracyclin, and quaternary ammonium compounds, among other chemicals[48]. Even though such resistance genes are not among the most relevant clinically, their presence is worth noting, given the scarcity of antibiotic resistance genes in phage genomes[49].

Beyond virulence factors and antibiotic resistance genes, temperate phages encode many other morons, defined as phage genes unnecessary for the phage lytic or lysogenic cycles, but believed to confer improved fitness to their host. Many of them have been characterized over the years (for a recent review, see[50]). We took advantage of our collection of 19 lambda to search for additional moron genes (Fig. 5a, see also lambda dotplots in Supplementary Fig. 8, and annotated alignments in Supplementary Fig. 9). An average of four predicted morons per lambda was found (counting the _sitABCD_ operon as a single moron). As prevalent as _sitABCD_ was the _bor_ gene, an outer membrane protein contributing to serum resistance. Other important contributors, present in more than 50% of lambdas, were _sie_, a superinfection exclusion gene and _blc_, a bacterial lipocalin, proposed to contribute to membrane biogenesis. Overall, a predominance of genes encoding putative membrane proteins, or membrane binding proteins, was found (bars tagged in yellow in Fig. 5a).

Lastly, a search for AMGs was conducted, building upon the list of 75 AMGs established based on large screenings of ocean and rumen viromes[51]. The _E. coli_ version of these metabolic genes was chosen to build up the protein subjects when available, or a close species otherwise. _E. coli_ homologs of reported phage encoded ribosomal proteins[52] were added to this list. A BLASTp search with E-value threshold $10^{-5}$ and bit-score threshold 50 revealed a reverse picture, relative to the search for virulence factors, whereby only virulent coliphages encoded AMG (Fig. 4 and Fig. 5b). Most AMGs were related to nucleotide metabolism, and our five rV5 encoded a ribosome hibernation factor (_hpf_ gene) as already described for the reference rV5[52].

**Receptor-binding proteins diversity.** Confronting the interaction matrix with genomic data revealed that closely related phages had different host ranges. In particular, a marked gradation of strain infectivity was observed for reference phage T5 (8% of strains infected) compared to our T5_ev219 (27% of strains

| Source | Temp / Vir | % Infection | Phage name (accession) | Length (bp) | Virfam classif[1] | Relative in Viruses at NCBI | Query cover (%) | Identity (%) | Resfam/VFDB/ AMG |
|---|---|---|---|---|---|---|---|---|---|
| Supnt | T | 5 | P2_4B2 (LR595873) | 32,118 | M 1, 9 | P2 | 74 | 98 | |
| Supnt | T | 7 | P2_2H4 (LR595869) | 31,342 | M 1, 9 | P2 | 83 | 98 | |
| Supnt | T | 2 | P2_4C9 (LR595883) | 32,150 | M 1, 9 | P2 | 70 | 97 | |
| Supnt | T | 0 | P2_4A7 (LR595887) | 32,848 | M 1, 9 | P2 | 74 | 96 | |
| Supnt | T | 1 | P2_4E6 (LR595885) | 31,128 | M 1, 9 | P2 | 71 | 96 | |
| Supnt | T | 4 | P2_2H1 (LR595870 ) | 32,662 | M 1, 9 | P2 | 79 | 98 | |
| Supnt | T | 8 | P88_4B11 (LR595891) | 37,100 | M 1, 9 | P88 | 80 | 98 | |
| Supnt | T | 1 | Mu_3F6 (LR595867) | 37,126 | M 1, 8 | Mu | 88 | 98 | |
| Supnt | T | 4 | mEp460_4F5 (LR595868) | 42,927 | S 1, 6 | mEp460 | 71 | 98 | |
| Supnt | T | 2 | Lambda_4C10 (LR595861) | 49,207 | S 1, 6 | Lambda | 30 | 96 | EmrE, SitABCD |
| Supnt | T | 0 | lambda_4A7 (LR595864) | 49,168 | S 1, 6 | Lambda | 31 | 96 | EmrE, SitABCD |
| Supnt | T | 4 | Lambda_2G7b (LR595866) | 47,142 | S 1, 6 | Lambda | 58 | 98 | SopA |
| Supnt | T | 0 | Lambda_2B8 (LR595859) | 48,763 | S 1, 6 | Lambda | 62 | 97 | EspM2 |
| Supnt | T | 1 | Lambda_2H10 (LR595862) | 46,288 | S 1, 6 | Lambda | 45 | 97 | |
| Supnt | T | 2 | Lambda_4B5 (LR595863) | 48,141 | S 1, 6 | Lambda | 52 | 97 | SitABCD |
| Supnt | T | 2 | Lambda_2G7a (LR595865) | 46,701 | S 1, 6 | Lambda | 54 | 97 | SitABCD |
| Supnt | T | 1 | Lambda_1H12 (LR595850) | 46,703 | S 1, 6 | Lambda | 54 | 97 | SitABCD |
| Supnt | T | 2 | Lambda_2E9 (LR595860) | 46,702 | S 1, 6 | Lambda | 54 | 97 | SitABCD |
| Virome | T | 0 | Lambda_ev203 (LR597650) | 46,680 | S 1, 6 | Lambda | 53 | 98 | SitABCD |
| Virome | T | 1,2 | Lambda_ev245 (LR597648) | 46,704 | S 1, 6 | Lambda | 54 | 98 | SitABCD |
| Virome | T | 3,5 | Lambda_ev110 (LR597652) | 46,770 | S 1, 6 | Lambda | 54 | 98 | SitABCD |
| Virome | T | 1,2 | Lambda_ev058 (LR597651) | 47,678 | S 1, 6 | Lambda | 53 | 98 | SitABCD |
| Virome | T | 3,5 | Lambda_ev087 (LR597644) | 46,710 | S 1, 6 | Lambda | 55 | 98 | SitABCD |
| Virome | T | 9,4 | Lambda_ev066 (LR597653) | 46,833 | S 1, 6 | Lambda | 54 | 98 | SitABCD |
| Virome | T | 3,5 | Lambda_ev243 (LR597639) | 45,560 | S 1, 6 | Lambda | 66 | 98 | SopA |
| Virome | T | 1,2 | Lambda_ev207 (LR597636) | 46,827 | S 1, 6 | Lambda | 58 | 97 | SitABCD |
| Virome | T | 2,4 | Lambda_ev099 (LR597635) | 47,332 | S 1, 6 | Lambda | 67 | 97 | |
| Virome | T | 1,2 | Lambda_ev017 (LR597643) | 50,126 | S 1, 6 | Lambda | 36 | 95 | SitABCD |
| Virome | T | 1,2 | mEp460_ev081 (LR597641) | 45 865 | S 1, 6 | mEp460 | 46 | 97 | |
| Virome | T | 1,2 | Evi (LR597642) | 40,019 | S 1, 3 | phi467 | 25 | 97 | EspV |
| Virome | T | 3,5 | Fraca (LR597645) | 43,664 | S 1, 3 | ECP1 | 29 | 94 | |
| Virome | T | 4,7 | ESSI2_ev239 (LR597637) | 29,203 | M 1, 9 | ESSI-2 | 31 | 77 | |
| Virome | T | 1,2 | ESSI2_ev129 (LR597640) | 30,927 | M 1, 9 | ESSI-2 | 34 | 77 | |
| Virome | T | 2,4 | ESSI2_ev015 (LR597649) | 30,584 | M 1, 9 | ESSI-2 | 31 | 77 | |
| Virome | T | 0 | ESSI2_ev040 (LR597638) | 30,597 | M 1, 9 | ESSI-2 | 31 | 77 | |
| Virome | V | 11,8 | Gluttony_ev152 (LR597646) | 44,825 | S 1, 5 | Gluttony | 89 | 93 | |
| Virome | V | 11,8 | Stevie_ev116 (LR597656) | 48,646 | S 1 | Stevie | 92 | 97 | |
| Virome | V | 12,9 | T4_ev240 (LR597638) | 166,954 | M 2 | RB32 | 95 | 97 | DHFR |
| Virome | V | 42 | T4_ev151 (LR597660) | 167,097 | M 2 | RB18 | 90 | 97 | DHFR |
| Virome | V | 15,3 | VpaE1_ev078 (LR597663) | 87,688 | M 1, 7 | VpaE1 | 94 | 97 | DHFR, Prs[2] |
| Virome | V | 29,4 | VpaE1_ev108 (LR597658) | 87,749 | M 1, 7 | VpaE1 | 94 | 96 | DHFR, Prs |
| virome | V | 10,6 | VpaE1_ev035 (LR699048) | 88,002 | M 1, 7 | VpaE1 | 93 | 96 | DHFR, Prs |
| Virome | V | 27,1 | T5_ev219 (LR597655) | 110,393 | S 1, 4 | T5 | 84 | 92 | |
| Virome | V | 40 | T5_ev212 (LR597659) | 108,546 | S 1, 4 | T5 | 89 | 92 | |
| Virome | V | 62,4 | phAPEC8_ev052 (LR597654) | 150,152 | M 1, 7 | phAPEC8 | 93 | 98 | RmlA, RmlB[3] |
| Virome | V | 23,5 | rV5_ev168 (LR694610) | 138,171 | M 1, 7 | rV5 | 96 | 97 | HPF |
| Virome | V | 42,4 | rV5_ev146 (LR699804) | 131,919 | M 1, 7 | rV5 | 94 | 98 | HPF |
| Virome | V | 60 | rV5_ev158 (LR694611) | 138,177 | M 1, 7 | rV5 | 96 | 97 | HPF |
| Virome | V | 66 | rV5_ev147 (LR597647) | 136,384 | M 1, 7 | rV5 | 97 | 97 | HPF |
| Virome | V | 68 | rV5_ev156 (LR694165) | 136,385 | M 1, 7 | rV5 | 97 | 97 | HPF |

**Fig. 4 Characteristics of the 50 sequenced coliphages genomes.** (1) M = myophage, S = siphophage. The first number denotes the Virfam Type. Second number is cluster number in Type 1. Each Virfam assignment is color coded as in the Virfam tree. The absence of any line between two phages indicates they belong to the same species (>95% id and >85% coverage with megablast). (2) Prs, Ribose-phosphate pyrophosphokinase. (3) RmlA, glucose-1-phosphate thymidylyltransferase; RmlB, dTDP-D-glucose 4,6-dehydratase.

infected) and T5_ev212 (40% of strains infected). Two encoded receptor-binding proteins have been reported in phage T5, allowing two different bacterial receptors to be recognized: FhuA and BtuB[53]. Moreover, T5 encodes various types of L shaped tail fibers, helping to overcome the bacterial surface shielding by O-glycosylated sugars. A comparison of the genome of two new T5-like phages with reference phages T5 and DT57C showed good overall synteny, except at two loci. One locus was at the receptor-

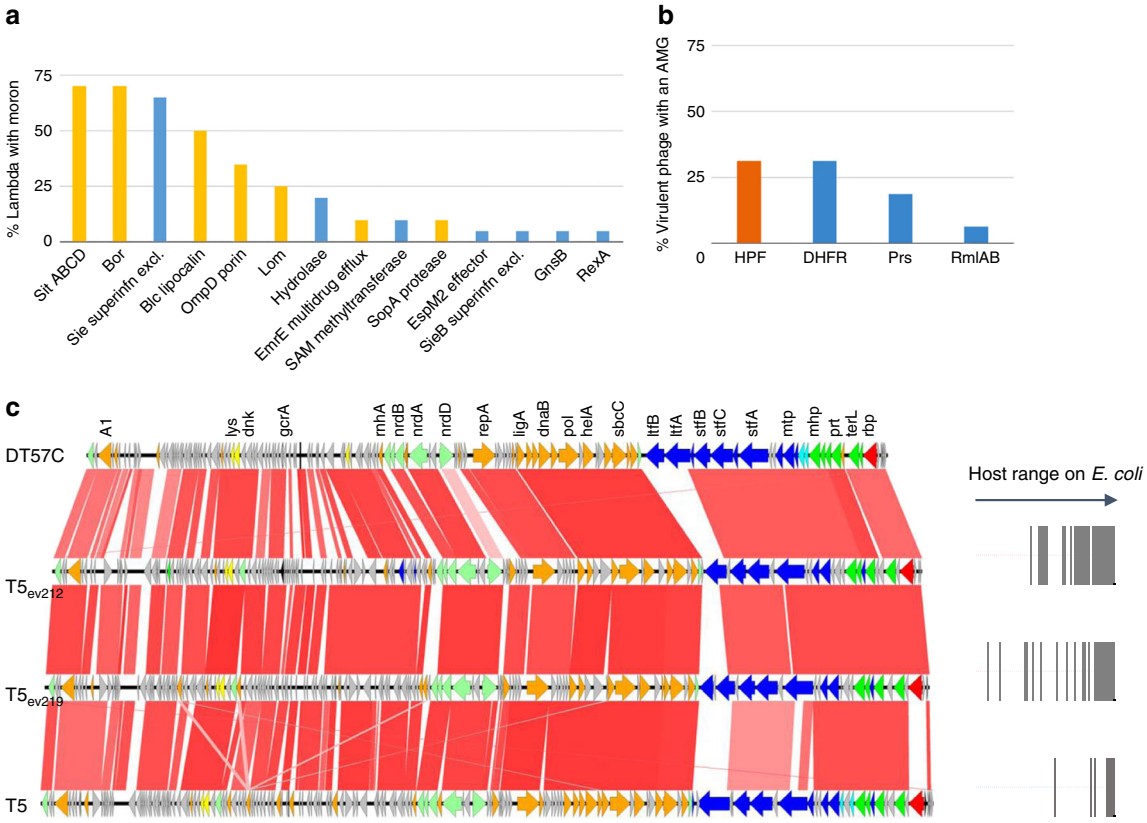

**Fig. 5 Salient features of coliphage genomes. a** Moron frequencies among 19 temperate Lambdaviruses. Yellow bars indicate membrane proteins and blue bars indicate other proteins. **b** AMG frequencies among 15 virulent sequenced phages. HPS, hibernation ribosomal protein; DHFR, dihydrofolate reductase; Prs, ribose-phosphate pyrophosphokinase; RmlA, glucose-1-phosphate thymidylyltransferase; RmlB, dTDP-glucose 4,6 dehydratase. **c** BLASTn alignment of four T5 genomes. The two T5-like phages isolated from viromes differ from T5 in their receptor-binding protein (p05 shown in red, similar to the DT57C gene), and from both DT57C and T5 in their L-shape tail fibers. The shade of red signals identity level (66–100%). Right panel: gray bars stand for strains on which each T5 grows (taken from Fig. 3).

binding protein, which was most similar to that of phage DT57C (73% AA identity) binding the BtuB bacterial receptor. The other clear interruption of synteny occurred at the position of the gene for the L-shape tail fiber proteins, which was different in each genome (Fig. 5c). This may explain the different host ranges (Fig. 5c, right part). A similar search for rV5 genomes did not reveal synteny breakpoints in the tail genes (Supplementary Fig. 10a). However, a closer look at polymorphism among the 18 tail proteins revealed that most were conserved, except two of them, Orf27 (annotated "structural protein with Ig domain, cell wall surface anchor" in the original rV5 genome) and Orf41 ("putative tail fiber protein"). Confronting the Neighbor-Joining tree of these two proteins with the host phage interaction matrix revealed an overall good correlation. One rV5 phage however, ev168 (Supplementary Fig. 10b, c), infected only 24% of the strains, whereas ev168, with very similar Orf27 and Orf41 proteins, infected 60% of the strains. This may be due to a truncated version of Orf38 ("tail assembly protein", the last 53 residues are lost) in ev168, preventing the correct arrangement of tail fibers in this phage (Supplementary Fig. 10d). Finally, among the lambda phages, four main clusters of the J adhesin protein were found, which did not match with their respective phage host range (Supplementary Fig. 11). Infection exclusion, or other host defense mechanisms may be at play for this phage family.

## Discussion

This large-scale coliphage study has enabled us to deepen our understanding of the impact of intestinal coliphages on their

hosts. An earlier study with feces samples from healthy adults reported that 90% of isolated coliphages were temperate[15]. Here, in 1-year old children, we also find a predominance of temperate coliphages (73%) in the fecal viromes. In addition, this study is the first, to our knowledge, to investigate coliphage growth on *E. coli* strains isolated from the same group of children. Even though they are more numerous relative to virulent phages, the impact of these temperate coliphages on *E. coli* isolates from 1-year old was low, with 0–9% (average 2%) of the tested strains infected by the 55 temperate phages found in strain supernatants or viromes. The inverse pattern was observed for the sub-dominant virulent phages that managed to infect 11–68% of the tested strains (average 32%).

While these virulent coliphages have the ability to replicate on several gut *E. coli* strains, it remains to be seen if they kill their bacterial hosts in vivo. For example, it is still unclear if the various phage receptors are expressed in the intestine, and if the bacteria themselves are accessible or shielded from phages in spatially heterogeneous and metabolically changing environments such as the human gut. Still, because they were readily isolated, these virulent phages are likely influencing the *E. coli* population in the gut of 1-year old children.

A main difficulty hindering the prediction of phage impact in the intestine is the absence of data on absolute phage titers in this environment, not to mention the phage to bacteria ratio for any given species. The most recent global estimates suggest average viral to microbial ratio around 0.1 in the gut[54]. Here, for *E. coli* in the infant gut, the highest estimated coliphage titers were $\sim 10^6$

PFU/g of feces (and thus may be 10-fold more, taking into account a possible decay during virome storage at 4 °C), and involved four virulent phages, namely Gluttony, a VpaE1, a T5 and an unsequenced phage (Fig. 2c, colored points). Their corresponding fecal samples were not depleted in *E. coli*, but rather tended to have higher relative levels of *E. coli* compared to the remaining samples (1–16%, Fig. 2b, colored points). Assuming a total number of bacteria per gr of feces of $10^{11}$, hence $10^9$–$10^{10}$ *E. coli*/g, leads for the best sample (VpaE1_ev078, and including the 4 °C decay factor), to estimate a maximal ratio of coliphages to *E. coli* of $7 \times 10^{-3}$. This ratio may be compatible with a sporadic bloom of these coliphages in the given child sample, thanks to an overall reasonable abundance of a sensitive *E. coli* host. Such abundance would facilitate the phage–host encounter along the virulent phage's travel through the gut, without drastically diminishing the total *E. coli* population. However, when such virulent phages enter imbalanced ecosystems dominated by *E. coli* such as those of infants or adults with diarrhea, they may multiply rapidly and perhaps facilitate a return to a lower, homeostatic *E. coli* population[15,16].

The two highest titers for the temperate phage present in viromes were ~$10^5$ PFU/g feces (so possibly up to 30 fold more, due to virome sample decay at 4 °C). Using the same assumptions as above (total number of bacteria per g of feces of $10^{11}$, hence 0.8 to $9 \times 10^9$ *E. coli*/g, inclusion of the maximal 4 °C decay factor), a maximal ratio of coliphages to *E. coli* of $3 \times 10^{-3}$ is estimated. In a study with germ-free mice colonized with two strains of *E. coli*, we found that the induction of prophage lambda was sufficiently high (1.6% per hour) to create a fitness cost[55]. In such a set-up, the phage-to-bacteria ratio was around $10^{-2}$, which is ~3 fold more than the highest ratio predicted here. Whether prophage induction is also a cost in complex environments remains to be investigated. We cannot rule out the idea that even though temperate phages are apparently unable to infect kins due to a lack of phage sensitive strains in the intestinal environment, they may impact the ecosystem by imposing fitness costs on their host.

What remains to be understood is the logic behind the maintenance of active temperate phages in an ecosystem such as the human gut, where they hardly manage to perform lytic cycles. One may speculate that the prophage-mediated "autolysis" function by itself is beneficial to the *E. coli* population as a whole. Such a situation has been observed with *Pseudomonas aeruginosa* during biofilm development, where prophage-mediated lysis serves for the late biofilm dissolution and colonization step[56]. A tripartite interaction protozoa-bacterium-prophage has also been proposed for the *stx*-encoding prophages, where prophage-mediated lysis liberates the Stx toxin in a sub-population, which then intoxicates some *E. coli* predator, although results are conflicting (see[57] and references therein). It is also possible, from the bacteriophage perspective, that these temperate phages cross-infect and multiply on other bacterial species. We suggested that some of our temperate coliphage isolates may in fact originate from *E. alberti* or Citrobacter (for ESSI2s), and from Enterobacter (for Fraca). Conversely, the first ESSI2 isolate to be described was reported to form clear plaques on its *Cronobacter sakazakii* host strain, despite its integrase gene[34]. This may suggest its incapacity to integrate, and possibly also its foreign bacterial host origin. A recent review focuses exactly on this point, namely how little we know of phage host ranges[58].

Among the 32 sequenced phages isolated from viromes, the most often encountered genus was temperate Lambdavirus (10 isolates). Among these lambda isolates, six were almost identical, and all but two encoded a SitABC-like iron transporter. This suggests relative success for these temperate phages, possibly related to the fact that it has genes allowing iron transport to their hosts. The next most frequently isolated phage clade was the

virulent *Vequintavirinae* subfamily of the *Myoviridae* family, with six independent isolates. These were also the phages with the broadest host range (av. 56%), suggesting a possible link between infectivity and prevalence for virulent (but not temperate) phages in infant gut. The reference phage V5 has raised interest due to its capacity to replicate on half of the tested pathogenic O157: H7 strains, and a quarter of the ECOR collection, as well as a large panel of pathogenic strains[59,60]. Moreover, several phA-PEC8 phages, which are distant relatives of rV5, are also reported to have broad host ranges[61]. Further work is needed to understand the molecular basis of such success observed for *Vequintavirinae* in *E. coli*. Nonetheless, it is tempting to speculate that a broad host range may be needed for virulent phages to persist in this fluctuating environment.

In conclusion, this study demonstrates that over 60% of the *E. coli* isolates from 1-old infants spontaneously release functional temperate phages. Considering that only two indicator strains were used, this already massive percentage might be even higher. However, these temperate phages have a narrow host range on *E. coli*. Virulent coliphages were also isolated from fecal samples but these distinct phages had a much larger host range, possibly to enable them to persist in this ecosystem. It remains to be seen if these findings can be generalized to other microbial species in the gut. This study highlights the interest of developing culturomic approaches to better understand the biological roles of phages in the intestinal microbiota.

## Methods

**Bacterial strains**. Two *E. coli* laboratory strains were used as indicator strains to screen coliphages. MAC1403 is an MG1655-derivative in which the KEIO allele *hsdR*:kanR was added by P1 transduction. Strain C is a natural *E. coli* isolate devoid of restriction activity (referenced under ATCC8739) and its genome is fully sequenced (NC_010468).

Natural *E. coli* strains were isolated from feces samples of 648 children that were included in the prospective Copenhagen Prospective Studies on Asthma in Childhood 2010 (COPSAC2010) mother-child cohort[62]. Among them, 900 strains came from one-year-old children fecal samples, 55 from 2-years-old children. Dilutions of the fecal samples were plated and grown aerobically. One to five colonies per sample, with different morphologies, were further cultivated and stored at −80 °C with 20% glycerol. Species determination was then determined biochemically, as already described[23].

Of these natural *E. coli* isolates, 75 were chosen to constitute a sub-panel for interaction analyses (70 from one-year-old children, and strains 9H12, 9A10, 8G12, 8C2, and 9G2 from two-years-old children). For this *E. coli* panel, a further classification was achieved using multiplex PCR, to identify the strains to the phylogroups A, B1, B2, C, D, E or cryptic Escherichia clades (named "E. clades")[29]. Briefly, a first quadruplex PCR reaction targeting three genes (*chuaA*, *yjaA*, and *arpA*) and a DNA fragment (TspE4.C2) was carried out with 1 μL of overnight culture. PCR products were then migrated on 2% agarose gel and exposed under UV light to reveal the isolate profile. The quadruplex genotype (presence/absence of the four PCR amplicons) was used to determine the phylogroup of each isolate. A second duplex PCR was done when the first quadruplex PCR was insufficient to discriminate between two phylogroups.

**Temperate coliphages from natural *E. coli* strains**. Natural *E. coli* isolates were cultivated in 96-well plates in LB medium. Following an overnight incubation at 37 °C, cultures were filtered through a 0.2 μm membrane filter (Pall AcroPrep Advance 96 filter plate). Supernatants were diluted in SM buffer (50 mM Tris pH 7.5, 100 mM NaCl, 10 mM MgSO4) at three different dilutions ($10^{-1}$, $10^{-2}$, $10^{-3}$). Next, double-layer plates with each bacterial indicator strain (MAC1403 or C) were inoculated with 5 μl aliquots of serially diluted phages using a Hamilton robotic system. Plates were then incubated overnight at 37 °C, and single plaque detection was used as an indicator of phage production. Each natural strain was tested twice on the two indicator strains. For strains where results were unclear, a third test was conducted. Phage concentrations in overnight cultures were estimated from the number of plaque forming units (PFU) detected at a given dilution. Among all phages obtained, 75 culture supernatants were randomly selected for further study (all of them from one-year-old children samples). Eighteen of them were plaque purified on either one of the two indicator strains, and amplified to produce larger stocks for DNA extraction and genome sequencing.

**Coliphage detection in fecal viromes**. We aimed to collect coliphages, both virulent and temperate, present in the fecal samples of 1-year-old children. The

protocol has been described in detail elsewhere [27], but in short, viromes were fractionated from 0.15 g of feces suspended into 29 ml 1× SM buffer (100 mM NaCl, 8 mM MgSO4•7H2O, 50 mM Tris-Cl with pH 7.5). The suspension was gently centrifuged (5000g for 30 min at 4 °C) to pellet larger particles and then filtered through 0.45 µm Minisart® High Flow PES syringe filter (Cat. No. 16533, Sartorius, Germany). The filtrate was concentrated using Centriprep® Ultracel® YM-50 K kDa units (Cat. No. 4310, Millipore, USA) until a final volume of 500 µL virome concentrate was obtained. From such viromes, 140 µL was taken for DNA extraction, and the rest was stored at 4 °C, and filtered through 0.2 µm Supor membrane (Acrodisc 13 mm syringe filter, PALL) prior usage for phage detection. To detect the presence of phages, 10 µl of each of the 648 virome samples were spotted on a double-layer plate inoculated with one of the two indicator strains, and incubated at 37 °C. The presence of confluent lysis or individual plaques was noted after overnight incubation. Within each 10 µL spot, in most cases there were less than 30 of them. In the few cases where lysis was confluent, dilutions were spotted in order to estimate the coliphage titer of the virome. The first 75 phages isolated were also purified twice by plaque streaking on the same indicator strain, amplified by confluent lysis on plates, and stored in SM buffer at 4 °C before being studied in greater detail.

**Coliphage host range determination**. Using the collection of 150 coliphages (half temperate and isolated from strains, half isolated from viromes) we determined which were able to grow on the panel of 75 natural *E. coli* strains. In addition, 16 virulent coliphages from the d'Hérelle collection were tested as a reference. To accomplish this, 1 mL of overnight culture was washed over the surface of a squared plate of dried LB supplemented with 5 mM CaCl2, 10 mM MgSO4 and 0.2% maltose, and the excess liquid was decanted. Then, 5 µl of filtered supernatants containing coliphages or purified coliphage stocks were applied on plates inoculated with the strain of interest. Plates were incubated overnight at 37 °C. The interactions were classified as either positive (with clear or turbid lysis plaques or confluent lysis spots) or negative (no lysis spots and no lysis plaques). The host-range determination was tested twice for each sample containing coliphages and if results were unclear, a third test was conducted.

**Analysis of interaction matrices with R package**. To investigate the modularity and nestedness of the interaction matrix, we removed from the dataset all the bacteria that were resistant to all phages ($n = 5/75$) and all the phages that were not infecting any bacteria ($n = 50/166$), resulting in a 70 bacteria × 116 phages interaction matrix. Modularity was quantified with the *lpbrim* package in R using the *findModules* function with 1000 iterations. The null model consisted of 1000 binomial matrices with the same number of interactions, but randomly distributed (Q value given in the matrix corner). Nestedness was measured with the *RInSp* package in R using the *NODF* function. Statistical significance was assessed with the *vegan* package using the *oecosimu* function with 1000 simulations (value indicated in the matrix corner).

**Relative abundance of *E. coli* in fecal samples**. In our earlier study, 16S rRNA amplicon sequencing of the microbiota of the same children had been performed[63]. To address the constant updates to the 16S databases, the proportion of *E. coli* 16S reads of the 1-year old infant fecal samples was tested again. For this, all operational taxonomic units (OTUs) formerly assigned to *Enterobacteria* were reanalyzed against the Silva database. This allowed adding two abundant *Enterobacteria* to the *E. coli* group.

**Coliphage genome sequencing**. For 50 of the coliphages isolated, DNA was extracted from crude phage stocks[64]. Crude lysates were PEG precipitated, followed by a DNaseI/RNase treatment to remove external contamination, and DNA was extracted with the Wizard DNA clean-up system (Promega). The detection of a DNA band for 5 µL of sample on an agarose gel, even if faint (~ above 2 ng), was considered sufficient to go for sequencing. Sequencing libraries were prepared using the Trueseq nano Sample Preparation kit, and samples were sequenced with an Illumina Miseq run. Read quality trimming was performed using Trimmomatic[65] and following options ILLUMINACLIP:Trueseq_nano.txt:2:30:10 LEADING:3 TRAILING:3 SLIDINGWINDOW:4:20 MINLEN:200. Trueseq nano kit tags were sometimes present inside reads, so they were systematically removed using the following sequences for clipping: CTGTCTCTTATACACATCTCCGAGCCC ACGAGAC for read 1 of the pair, and TGTAGATCTCGGTGGTCGCCGTA TCATTAAA for read2. Genome assembly was performed with spades v3.13.0[66] using only paired reads, and following options—only-assembler—meta -k 21,33,55,77,99,127. Phage genome was considered complete when spades returned a single contig with high coverage. Repeats at the contig ends were removed prior choosing genome orientation and nt 1 position, relative to reference genomes. In three cases where spades returned more than one contig with high coverage, assembly was completed manually, based on overlapping sequences at contig ends.

**Dotplot analyses, genome annotations, and comparisons**. Assembled phage genomes were compared to the NCBI Viruses section database using megablast to choose a well-known reference genome for each genus (indicated in Table 2). Dotplot graphs were generated using Gepard[67] (word size = 10). Automatic ORF calling and annotation were performed using RAST[68] and the Virus genome

option. Manual editing was done for the connector genes detected with Virfam, as well as for lambda morons. Reference phage phAPEC8 being poorly annotated, its annotation was completed using distant homology searches, thanks to Phagonaute interface[69]. The T5s and Mu automatic annotations were completed following DT57C and Mu genbank files. tRNA were searched with tRNAscanSE 2.0[70]. A systematic search for reverse transcriptases, frequently found in intestinal phages[18], was conducted with hmmscan against Pfam profile PF00078, and none was found. Antibiotic resistance genes were searched using ResFam core profiles and hmmscan as indicated by the authors[47]. All virulence factors detected thanks to the Virulence Factor Database had been correctly assigned by RAST, except for the *espM2* and *espV* genes. All genomes were deposited at EBI (accession numbers in Fig. 4).

Alignments and result displays were performed with Easyfig[71] (BLASTn), starting from the Genbank files in which a field "/color" was added to help recognize phage modules: orange for replication/recombination genes, green for capsids, light blue for connectors, dark blue for tails and tail fiber genes, yellow for lysis genes, red for integrase (unless otherwise stated), black for morons and gray for genes of unknown function.

**Ethical compliance**. The study was performed according to the principles of the Declaration of Helsinki and was approved by the Ethics Committee of Copenhagen (H-B-2008-093) and the Danish Data Protection Agency (2015-41-3696). We are aware of and comply with recognized codes of good research practice, including the Danish Code of Conduct for Research Integrity. We comply with national and international rules on the safety and rights of patients and healthy subjects, including Good Clinical Practice, as defined in the European Union's Directive on Good Clinical Practice, the International Conference on Harmonisation's good clinical practice guidelines, and the Helsinki Declaration. We follow national and international rules on the processing of personal data, including the Danish Act on Processing of Personal Data and the practice of the Danish Data Inspectorate. Mothers gave informed consent for their child.

**Reporting summary**. Further information on research design is available in the Nature Research Reporting Summary linked to this article.

## Data availability
All strains and phages described in the study are available upon request. The source data underlying Fig. 2, Supplementary Figures S1, S2, S3 and accessions numbers of genomes described in Fig. 4 are provided as a Source Data file.

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

## Acknowledgements

This work is supported by the Joint Programming Initiative 'Healthy Diet for a Healthy Life'. The funding agencies supporting this work are: The Danish Agency for Science and

Higher Education, Institut National de la Recherche Agronomique et Environnementale (INRAE) and the Canadian Institutes of Health Research. M.B.D. is recipient of a graduate scholarship from the Fonds de Recherche du Québec—Nature et Technologies. S.M. holds the Tier 1 Canada Research Chair in Bacteriophages. We express our deepest gratitude to the children and families of the COPSAC2010 cohort study for all their support and commitment.

## Author contributions

H.B. and J.S. built up the COPSAC cohort and provided fecal samples. K.A.K. and S.S. curated the E. coli collection and provided strains. L.D. and D.S.N. prepared the viromes. A.M. isolated all phages and performed interaction experiments with the help of E.M. M.D., D.T. and S.M. contributed to phage sequencing and interaction studies. S.A.S. and M.A.P. contributed to bio-informatics studies. A.M., M.D., S.M. and M.A.P. wrote down the manuscript, and all authors improved and approved the final version of the manuscript.

## Competing interests

The authors declare no competing interests.
