## [Peer Review File · Nature Communications]

Reviewers' comments:

Reviewer #1 (Remarks to the Author):

In this manuscript Mathieu et al. perform an in depth characterization of numerous coliphages, both lytic and temperate, isolated from the feces of 1 year old infants. They screen these phages for host range on a large cohort of E. coli isolates from the same infants. They discovered that greater than half of the isolates carried integrated prophages and that these temperate phages had rather rigid host tropisms, which do not seem to be associated with lysogeny. Their data indicate that genomic diversity possibly mediated by phage tail and bacterial receptor incompatibilities drive this high level of resistance. All together this is a well written manuscript that provides a detailed atlas of intestinal coliphages and their narrow host spectrum. This work provides important insight into a possible mechanism for the observed inter-individual variation in intestinal viromes and why some phages might become dominant within an individual.

I have a few specific comments for the authors moving forward. This is largely related to going deeper with their analysis to reveal potential novelties that have otherwise been under-described for intestinal coliphages.

Major comments

1. It is striking that many of the lysogenized strains are resistant to infection by other temperate phages considering at the genome comparison level, all of these phages group into 3 distinct phage types. Their data suggest that this is not due to superinfection exclusion by previously lysogenized cells, indicating that host range is dictated based on phage/bacterial receptor compatibility. They hint at this by doing a comparison of two phage tail proteins, showing they have variation that could account for this.

First, what about a broader comparison of tail proteins among the 3 distinct phage types. Do the phage tail sequences group in any meaningful way that could provide insight into host tropism patterns?

Second, what about doing genetic swaps to test this theory? Presumably they know which phages are lysogenized in which hosts and since these are integrated prophages, doing genetics to change out some of the tail proteins to show altered tropism should be feasible.

2. The really fascinating aspect of this work in my opinion relates to the categories of accessory genes that were discovered in many of these coliphages. However, this is only verified based on genome annotation analysis. Since there are testable phenotypes predicted to be associated with some of these accessory genes (i.e. metal transport, antibiotic resistance, etc.) and the infection of susceptible cells by these temperate phages presumably results in lysogeny based on the hazy plaque phenotype, one could test for gain of function for any one of these interesting phenotypes to show the importance of one or more accessory genes.

Minor comments

1. In Fig. 2B, why are some of the virulent phage data points a different color? This is not indicated in the figure legend.

2. In Fig. S2, I recommend plotting the data on an arithmetic scale since this data represents an absolute abundance change at a distinct time point.

3. It is not clear in the text how E (N=14) and E.clades differ, as shown in Fig. S3A.

Reviewer #2 (Remarks to the Author):

The work is well done and clearly described; I have no requests for revision, besides the following quite minor points:

1) the legend of Figure 1, A mentions '2-year-old infants', whereas the rest of the paper is written about 1-year-old children;

2) Reference #38 in Table 1 is likely incorrect.

We would like to mention to both referees that the number of E. coli strains from 1-year old children that we tested was 900, and not 954 as wrongly stated all over the manuscript. We learned a posteriori that the 54 extra strains had been isolated from 2-year old children fecal samples, so we removed them from the analysis. This change did not impact the rest of the results, the 954 number has been replaced by 900 all over the manuscript.

Reviewer #1 (Remarks to the Author):

Major comments

1. It is striking that many of the lysogenized strains are resistant to infection by other temperate phages considering at the genome comparison level, all of these phages group into 3 distinct phage types. Their data suggest that this is not due to superinfection exclusion by previously lysogenized cells, indicating that host range is dictated based on phage/bacterial receptor compatibility. They hint at this by doing a comparison of two phage tail proteins, showing they have variation that could account for this.

First, what about a broader comparison of tail proteins among the 3 distinct phage types. Do the phage tail sequences group in any meaningful way that could provide insight into host tropism patterns?

Thank you for suggesting this. We further analyzed the tail proteins of additional phage groups, and added the following :

-with respect to rV5, the tail fiber diversity was not as striking as for T5 (lines 381-389 and new suppl Fig10). However, a systematic search for polymorphism among the 27 tail proteins (dark blue genes in panel a) revealed for two of them a nice polymorphism in line with host range (panels b and c). Only one phage (rV5_ev168) exhibited incongruity between host range and these tail fibers phylogenies. We found out that the tail assembly protein orf38 was truncated in this phage (panel d). This may explain its relative poor host range (24%), compared to other rV5 (42-66%).

-with respect to Lambda genomes, the J proteins group into 4 clusters that do not enlight the host tropism patterns (new Suppl Fig 11 and lines 389-391). So for this taxon, we cannot link host tropism pattern with tail polymorphism. Other phage resistance mechanism(s) is probably at play.

Second, what about doing genetic swaps to test this theory? Presumably they know which phages are lysogenized in which hosts and since these are integrated prophages, doing genetics to change out some of the tail proteins to show altered tropism should be feasible.

We already performed C-ter swapping of Lambda J proteins in the past, and found this was not as trivial as one could imagine. See also this recent paper:

<https://www.sciencedirect.com/science/article/pii/S2211124719312598?via%3Dihub>

So we prefer to keep this interesting question for future explorations.

2. The really fascinating aspect of this work in my opinion relates to the categories of accessory genes

that were discovered in many of these coliphages. However, this is only verified based on genome annotation analysis. Since there are testable phenotypes predicted to be associated with some of these accessory genes (i.e. metal transport, antibiotic resistance, etc.) and the infection of susceptible cells by these temperate phages presumably results in lysogeny based on the hazy plaquing phenotype, one could test for gain of function for any one of these interesting phenotypes to show the importance of one or more accessory genes.

We agree that these accessory genes of temperate phages are an important piece of the puzzle. While the complete testing of their associated phenotypes is beyond the reach of this publication, we sought to strengthen the evidence that the prevalent *sitABCD* operon is functional in the 14/20 Lambda in which it was found. We searched for *fur* boxes, which would indicate gene expression is under the control of iron homeostasis master regulator Fur, and found them upstream of the operon in all cases. This is now mentioned lines 319-320 and added as Suppl. Fig 9.

Minor comments

1. In Fig. 2B, why are some of the virulent phage data points a different color? This is not indicated in the figure legend.

A different color is given in panel B for the temperate and the virulent phage isolated from viromes. This is now stated in the figure legend, lines 201-203.

2. In Fig. S2, I recommend plotting the data on an arithmetic scale since this data represents an absolute abundance change at a distinct time point.

Done

3. It is not clear in the text how E (N=14) and E.clades differ, as shown in Fig. S3A.

“E” stands for phylotype E, and “E. clade” stands for cryptic *Escherichia* clades that cannot be typed with the multiplex PCR. We now clarify this better in the text, lines 491-492.

Reviewer #2 (Remarks to the Author):

The work is well done and clearly described; I have no requests for revision, besides the following quite minor points:

1) the legend of Figure 1, A mentions '2-year-old infants', whereas the rest of the paper is written about 1-year-old children;

Corrected (line 114)

2) Reference #38 in Table 1 is likely incorrect.

Corrected (line 255)

REVIEWERS' COMMENTS:

Reviewer #1 (Remarks to the Author):

The authors have done a mostly adequate job addressing my previous comments. However, there is still one sticking point that remains to be addressed.

This is in response to the testing of the phage tail protein specificity by performing genetic swapping experiments. Since the overall impact of this work is based on the idea that these temperate coliphages have limited tropism due to what the authors conclude is phage/bacterial receptor compatibility and not super infection exclusion as one would presume, then a genetic swapping experiment would be paramount to such a conclusion.